# Pre-operative stereotactic radiosurgery and peri-operative dexamethasone for resectable brain metastases: a two-arm pilot study evaluating clinical outcomes and immunological correlates

Enhancing the efficacy of immunotherapy in brain metastases (BrM) requires an improved understanding of the immune composition of BrM and how this is affected by radiation and dexamethasone. Our two-arm pilot study (NCT04895592) allocated 26 patients with BrM to either low (Arm A) or high (Arm B) dose peri-operative dexamethasone followed by pre-operative stereotactic radiosurgery (pSRS) and resection ($n$ = 13 per arm). The primary endpoint, a safety analysis at 4 months, was met. The secondary clinical endpoints of overall survival, distant brain failure, leptomeningeal disease and local recurrence at 12-months were 66%, 37.3%, 6%, and 0% respectively and were not significantly different between arms ($p$ = 0.7739, $p$ = 0.3884, $p$ = 0.3469). Immunological data from two large retrospective BrM datasets and confirmed by correlates from both arms of this pSRS prospective trial revealed that BrM CD8 T cells were composed of predominantly PD1+ TCF1+ stem-like and PD1+ TCF1-TIM3+ effector-like cells. Clustering of TCF1+ CD8 T cells with antigen presenting cells in immune niches was prognostic for local control, even without pSRS. Following pSRS, CD8 T cell and immune niche density were transiently reduced compared to untreated BrM, followed by a rebound 6+ days post pSRS with an increased frequency of TCF1- effector-like cells. In sum, pSRS is safe and therapeutically beneficial, and these data provide a framework for how pSRS may be leveraged to maximize intracranial CD8 T cell responses.

CD8 T cell infiltration in primary and metastatic malignancies is associated with improved outcomes and superior responses to immunotherapy[1–4]. Studies of the CD8 T cell response to tumors have highlighted the importance of a TCF1+ stem-like subset. This cell self-renews and gives rise to cytotoxic effector daughter cells in the setting of chronic antigen exposure[5] and across several human tumor types[6–10]. Functional experiments and TCR sequencing analysis indicate stem-like CD8 T cells retain proliferative capacity and are likely the source of anti-tumor effector CD8 T cells[8,11,12]. Our prior work showed these cells reside in close proximity to densely clustered MHC-II+ antigen-presenting cells (APCs) in the tumor, which we termed antigen-presenting niches, or immune niches[13]. We also recently

✉e-mail: haydn.kissick@emory.edu; zbuchwa@emory.edu

reported that these stem-like TCF1+ CD8 T cells in head and neck tumors are specific for HPV antigens, and upon restimulation with their cognate antigen, proliferate and differentiate to the effector state[8]. Furthermore, numerous studies correlate the presence of this CD8 T cell population in the tumor with response to PD-1 blockade, high-lighting their importance in the immune response to cancer[10,14,15].

Brain metastases (BrM), despite residing in an immune-privileged organ, have notable immunologic similarities to other metastatic sites, including the presence of antigen-experienced TCF1+ CD8 T cells[16–19]. A recent study also suggested that these cells may have overlapping TCR clonotypes with a more exhausted pool of TOX+ CD8 T cells present in BrM[19]. Furthermore, the specific spatial organization of BrM immune infiltrates appears to influence prognosis[20]. However, whether the TCF1+ CD8 T cell population is organized into antigen-presenting niches and whether this also has prognostic value is unknown.

Importantly, for patients with a limited number of large or symptomatic BrM, surgery followed by stereotactic radiosurgery (SRS) (post-operative) is frequently recommended[21,22]. Our group has been a pioneer in the pre-operative delivery of SRS[23]. Given the known immunostimulatory activity of radiation, pre-operative SRS (pSRS) may potentiate the intracranial activity of immunotherapy[24,25]. In this work, we performed both a retrospective analysis and completed a prospective clinical trial evaluating the safety, efficacy, and immunomodulatory activity of pSRS and peri-operative steroid dosing. We found that pSRS can be delivered safely, and that dexamethasone dose did not have a significant impact on clinical outcomes. For the immunological correlates, we evaluated the organization of TCF1+ PD1+ T cells in BrM under baseline conditions and the association of this biology with patient outcomes finding that increased BrM niche density was associated with longer local BrM control. We also assessed the impact of pSRS on the intracranial CD8 T cell immune response, finding that pSRS is immunomodulatory, altering both the T cell and APC compartments, while the overall immune organization is maintained.

## Results

### Study design and patient disposition

To prospectively investigate the impact of both pre-operative SRS (pSRS) and dexamethasone on brain metastases (BrM) clinical outcomes and CD8 T cell/APC subsets, a total of 36 patients with BrM were screened and 26 were enrolled on a pSRS clinical trial between August 2021 and April 2023 (Fig. 1A, B). Due to common dexamethasone administration for patients with BrM and unknown effects on clinical outcomes and the tumor immune microenvironment, enrollees were allocated to Arm A (≤ 4 mg of peri-operative dexamethasone daily) or Arm B (≥ 16 mg daily) in an alternating fashion, followed by pSRS and surgical resection for both arms. The characteristics for the 26 patients enrolled are shown by Arm in Supplementary Table 1. A total of 21 patients completed all therapy and were included in our analysis (Table 1).

### Primary and Secondary Outcomes

For these 21 patients, there were 3 deaths in Arm A prior to the final study visit, 2 in Arm B, and 1 patient in Arm B was unable to complete the final study visit leading to a final total of 15 patients reaching the pre-specified safety analysis endpoint of 4 months (Fig. 1B). The median follow-up time was 10.32 months (range: 0.84-23.04). Toxicities for the entire treated cohort are summarized in Table 2, with only two grade 3 or greater toxicities observed and attributable to therapy (altered mental status and cerebral edema, both in the same patient on Arm B) during the trial. Clinical endpoints including the cumulative incidence of local recurrence, distant brain failure (DBF), leptomeningeal disease (LMD), and overall survival (OS) were also assessed. The 12-month local recurrence was 0%. The 12-month cumulative incidence of DBF and LMD competing with death were 37.3% and 6.0%, respectively (Fig. 1C). The 12-month cumulative incidence of DBF was

23.1% for Arm A and 49.1% for Arm B (Gray test $p$= 0.3884; 95% CI: 0.033-0.533) (Fig. 1D). The 12-month cumulative incidence of LMD was 0.0% for Arm A and 11.4% for Arm B (Gray test $p$= 0.3469; 95% CI: N/A-0.394) (Fig. 1D). The 12-month overall survival was 66.0% (Fig. 1C). The 12-month overall survival was 80.5% for Arm A and 49.7% for Arm B ($p$= 0.7739; 95% CI: 0.20-3.33) (Fig. 1D).

This trial met its primary safety endpoint. Additionally, the secondary clinical outcomes, similar to results seen on other retrospective cohort studies for pre-operative SRS[26], compare favorably in both safety and outcomes to historic controls of upfront resection[23].

### Exploratory Immunologic Outcomes with Comparison Datasets

**TCF1+ PD1+ stem-like T cells are found in brain metastases and reside in immunological niches.** To characterize the CD8 T cell response in untreated BrM, we first performed flow cytometry analysis on freshly resected tissue from six patients in an independent cohort not treated on our prospective trial (Fig. 2A, Supplementary Fig. 1A). The total CD8 T cell infiltration was variable across patients (Supplementary Fig. 1B), with most tumor infiltrating CD8 T cells expressing PD1 and negative for CD45RA, indicative of antigen reactivity (Fig. 2B). Among PD1 + CD8 T cells, we identified an effector-like population expressing inhibitory molecules (TIM3 and CD39), as well as the residency marker CD69 (Fig. 2B, C). In contrast, a second CD8 T cell population, which made up around 30% of the PD1+ response, was negative for effector and inhibitory molecules but retained high expression of the transcription factor TCF1 in BrM across patients (Fig. 2B, C). Furthermore, these cells also expressed high levels of the co-stimulatory molecule CD28, and the IL-7 receptor compared to TCF1- effector-like CD8 T cells (Fig. 2B, C). As the phenotype of this population is consistent with previously defined stem-like CD8 T cells in other tumor types outside of the CNS and chronic infections[5,7,12], we also refer to these cells as "stem-like".

To further characterize and acquire spatial information of the BrM immune landscape, we evaluated 67 up-front resected samples at Winship Cancer Institute of Emory University and performed quantitative multiparametric immunofluorescence (Fig. 2D, E, Supplementary Tables 2-3). This retrospective cohort of BrM, which were not treated on our prospective study, underwent upfront resection (Res) and included non-small cell lung cancer (NSCLC), melanoma, or breast cancer; none of which had previously received immunotherapy (Supplementary Table 2). We found a high degree of variation between patients in the infiltration of CD8 and CD4 T cells, as well as of MHC-II+ antigen presenting cells (APCs) (Fig. 2E). Consistent with the flow cytometry results, the majority of CD8 T cells expressed PD1 (95%) and were found throughout the tumor tissue (Supplementary Fig. 1C–F). Additionally, approximately 40% of the total CD8 T cell population were TCF1+ (stem-like) and the remainder TCF1- cells (effector-like) (Supplementary Fig. 1F, G, Fig. 2E). It is worth noting that the frequency of TCF1+ CD8 T cells (of DAPI+ cells) correlated with the frequency of total CD8 T cells ($R^2$ = 0.6233, p < 0.001) suggesting that this population could sustain the overall CD8 T cell response in BrM (Supplementary Fig. 1H).

In our prior work, we found that stem-like TCF1+ CD8 T cells were not randomly distributed in kidney tumors, but instead resided in densely clustered APC immune niches[7]. Here, we found a correlation between the density of TCF1+ CD8 T cells and MHC-II+ APCs (Supplementary Fig. 1H) in BrM, suggesting a similar relationship. To quantitatively assess the proximity of CD8 T cell populations with APCs, we generated contour maps of the density of MHC-II+ cells and overlayed the x, y location of TCF1+ CD8 T cells (green) and of TCF1- CD8 T cells (red) (Fig. 2F). We found that TCF1+ CD8 T cells resided in areas of higher MHC-II+ cell density (Fig. 2F), and on average were significantly closer to their nearest MHC-II+ cell neighbor than TCF1- CD8 T cells (Fig. 2G). In contrast, TCF1- CD8 T cells were distributed throughout the tumor without any preference for APC zones, consistent with the biology of PD1+ TCF1+ CD8 T cells specifically requiring these

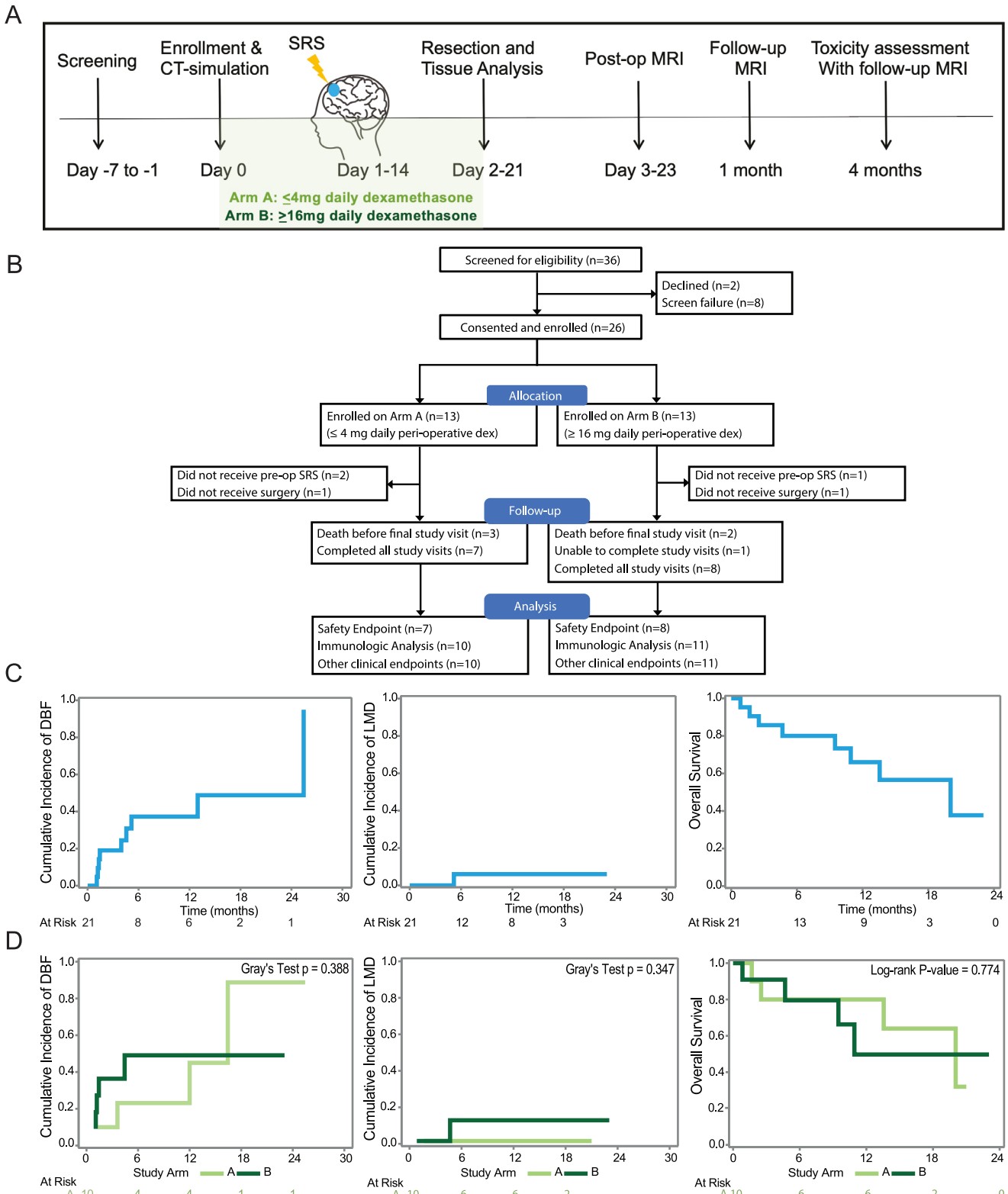

**Fig. 1 | pSRS can be delivered safely and effectively. A** Clinical trial schema. **B** Consort diagram. **C** Cumulative incidence curves for distant brain failure (DBF), leptomeningeal disease (LMD), and overall survival (OS) all analyzed patients. **D** Cumulative incidence for DBF, LMD, and OS for patients assigned to Arm A and Arm B. The *p*-value is calculated by Gray's test and Log-rank test. All tests are two-sided. Source data are provided as a Source Data file.

organized immune niches within the tumor. Prior data suggested that immune niches were important to sustain the CD8 T cell response within non-CNS tumors[13]. Notably, our BrM cohort showed a correlation between immune niche density and the frequency of total CD8 T cell infiltration (Fig. 2H), suggesting a similar supportive role for immune niches in BrM (Supplementary Fig. 2A–F).

Although our patient cohort was comprised of a diverse set of primary tumor types, we did not find significant differences in number

**Table 1 | Patient Characteristics on Prospective Trial**

| Covariate | Study Arm | | |
|---|---|---|---|
| | A (N = 10) | B (N = 11) | P-value |
| **Age** | 63.05 ± 7.91 | 57.53 ± 12.85 | 0.257 |
| **Gender** | | | |
| Female | 5 (50) | 7 (63.64) | 0.670 |
| Male | 5 (50) | 4 (36.36) | |
| **ECOG** | | | |
| <2 | 9 (90) | 7 (63.64) | 0.311 |
| >=2 | 1 (10) | 4 (36.36) | |
| **Primary Site** | | | |
| NSCLC | 3 (30) | 2 (18.18) | 0.893 |
| Breast | 1 (10) | 3 (27.27) | |
| Melanoma | 3 (30) | 4 (36.36) | |
| GI | 1 (10) | 1 (9.09) | |
| Other | 2 (20) | 1 (9.09) | |
| **Previous IO therapy** | | | |
| No | 7 (70) | 9 (81.82) | 0.635 |
| Yes | 3 (30) | 2 (18.18) | |
| **Preop SRS dose (Gy)** | | | |
| 12 | 0 (0) | 1 (9.09) | 0.601 |
| 13 | 1 (10) | 2 (18.18) | |
| 14 | 2 (20) | 1 (9.09) | |
| 16 | 2 (20) | 2 (18.18) | |
| 18 | 3 (30) | 1 (9.09) | |
| 24 | 0 (0) | 3 (27.27) | |
| 27 | 2 (20) | 1 (9.09) | |
| **Preop SRS fractions** | | | |
| 1 | 8 (80) | 7 (63.64) | 0.635 |
| 3 | 2 (20) | 4 (36.36) | |
| **WBC** | 9.18 ± 4.52 | 11.2 ± 5.69 | 0.382 |
| **ANC** | 7.37 ± 3.94 | 8.39 ± 4.88 | 0.608 |
| **Lymphocytes** | 1.49 ± 0.68 | 1.76 ± 2.1 | 0.706 |
| **SRS GTV volume (cm³)** | 17.3 ± 14.58 | 29.44 ± 18.26 | 0.111 |
| **Days from SRS to surgery** | 3.5 ± 3.31 | 4.55 ± 2.94 | 0.453 |

The p-value is calculated by two-way ANOVA for numerical covariates; and chi-square test or Fisher's exact for categorical covariates, where appropriate. Values for categorical factors are presented as numbers (percentage), and for continuous variables are presented as mean ± standard deviation. All tests are two-sided.

**Table 2 | Adverse Events within 4 months**

| Adverse Event | Arm A (n = 10) | | Arm B (n = 11) | |
|---|---|---|---|---|
| | Grade 1-2 (%) | Grade 3-4 (%) | Grade 1-2 (%) | Grade 3-4 (%) |
| Altered Mental Status | 0 | 0 | 0 | 1 (9.1) |
| Cerebral Edema | 0 | 0 | 0 | 1 (9.1) |
| Cognitive Disturbance | 1 (10.0) | 0 | 0 | 0 |
| Confusion | 5 (50.0) | 0 | 1 (9.1) | 0 |
| Dysarthria | 1 (10.0) | 0 | 0 | 0 |
| Dysphasia | 0 | 0 | 1 (9.1) | 0 |
| Facial Muscle Weakness | 1 (10.0) | 0 | 0 | 0 |
| Fatigue | 2 (20.0) | 0 | 3 (27.3) | 0 |
| Gait Abnormality | 0 | 0 | 1 (9.1) | 0 |
| Generalized Muscle Weakness | 1 (10.0) | 0 | 1 (9.1) | 0 |
| Headache | 3 (30.0) | 0 | 2 (18.2) | 0 |
| Paresthesia | 0 | 0 | 1 (9.1) | 0 |
| Radiation Necrosis | 0 | 0 | 1 (9.1) | 0 |
| Scalp Pain | 1 (10.0) | 0 | 0 | 0 |
| Somnolence | 2 (20.0) | 0 | 0 | 0 |
| Symptomatic | 0 | 0 | 1 (9.1) | 0 |
| Tremor | 0 | 0 | 1 (9.1) | 0 |
| Vision Changes | 1 (10.0) | 0 | 3 (27.3) | 0 |

of total CD8 T cells, TCF1+ , TCF1- CD8 T cell subsets/mm² or niche density by tumor histology (Supplementary Table 3, Supplementary Fig. 2G, K). Moreover, we did not find any significant association between dexamethasone dose and CD8 cells per mm², TCF1+ frequency or immune niche proportion, despite the known effects of glucocorticoids on CD8 T cells[27,28] (Supplementary Table 3). Overall, these data indicate that CD8 T cell responses are successfully mounted in BrM, despite being present in an immune privileged site. Moreover, these PD1+ TCF1+ CD8 T cells reside within APC-dense immune niches, which are maintained in the presence of dexamethasone.

**High immune niche density is associated with longer BrM local control.** CD8 T cell infiltration and a high density of local immune niches have previously been associated with improved clinical outcomes in solid tumors[1-4]. Thus, we next examined whether high CD8 T cell density or immune niches are associated with a longer time to BrM local recurrence in our retrospective dataset. We performed a competing risk analysis in a patient cohort that did not receive pSRS,

allowing us to account for the competing events of death and local recurrence. Although total CD8 T cell density was not associated with disease progression as previously seen for other solid malignancies (Supplementary Fig. 2L), BrM tumors with a greater frequency of PD1+ TCF1+ CD8 T cells or a higher density of immune niches were significantly associated with longer time to local recurrence (Fig. 2I). A representative patient with a high density of TCF1+ CD8 T cells and immune niches distributed throughout the BrM is highlighted, where the resection cavity remained free of local recurrence at ten years (Fig. 2J). In contrast, a representative patient with low TCF1+ CD8 T cell infiltration and scant immune niche formation developed local recurrence within six months after treatment (Fig. 2K). Together, these data suggest that the formation of immune niches appears to capture a link between the immune microenvironment and patient outcomes. These favorable responses are not evident with bulk CD8 T cell analysis alone, highlighting the importance of stem-like CD8 T cells and immune niche organization in restraining tumor growth and local recurrence.

**The impact of pre-operative SRS on T cell subsets and the immune niche.** In preclinical studies, focal radiation stimulates innate immune responses via the c-GAS-STING pathway and increases type I interferon signaling[29,30]. Additionally, we previously found focal radiation increased stem-like CD8 T cell infiltration into the tumor in murine cancer models[25]. However, whether pSRS has similar immunostimulatory activity in the brain and an analogous impact on the BrM immune response in patients is not known. To examine the impact of pSRS on BrM, we performed quantitative IF on 76 patients who were treated with pSRS 0-11 days prior to surgery from our retrospective dataset. Relevant patient, BrM, and pSRS characteristics are detailed in Supplementary Table 4. As a comparison, we included the retrospective upfront resection cohort (Res) for this analysis. Importantly, the pSRS and Res cohort characteristics were overall very similar, including in BrM volume (Fig. 3A, Supplementary Table 5).

The pSRS BrM had a wide range of T infiltration, with CD4 and CD8 T cells ranging from 8 to over 100 cells per mm² of tissue, which was significantly lower than Res BrM tumors (Fig. 3B and

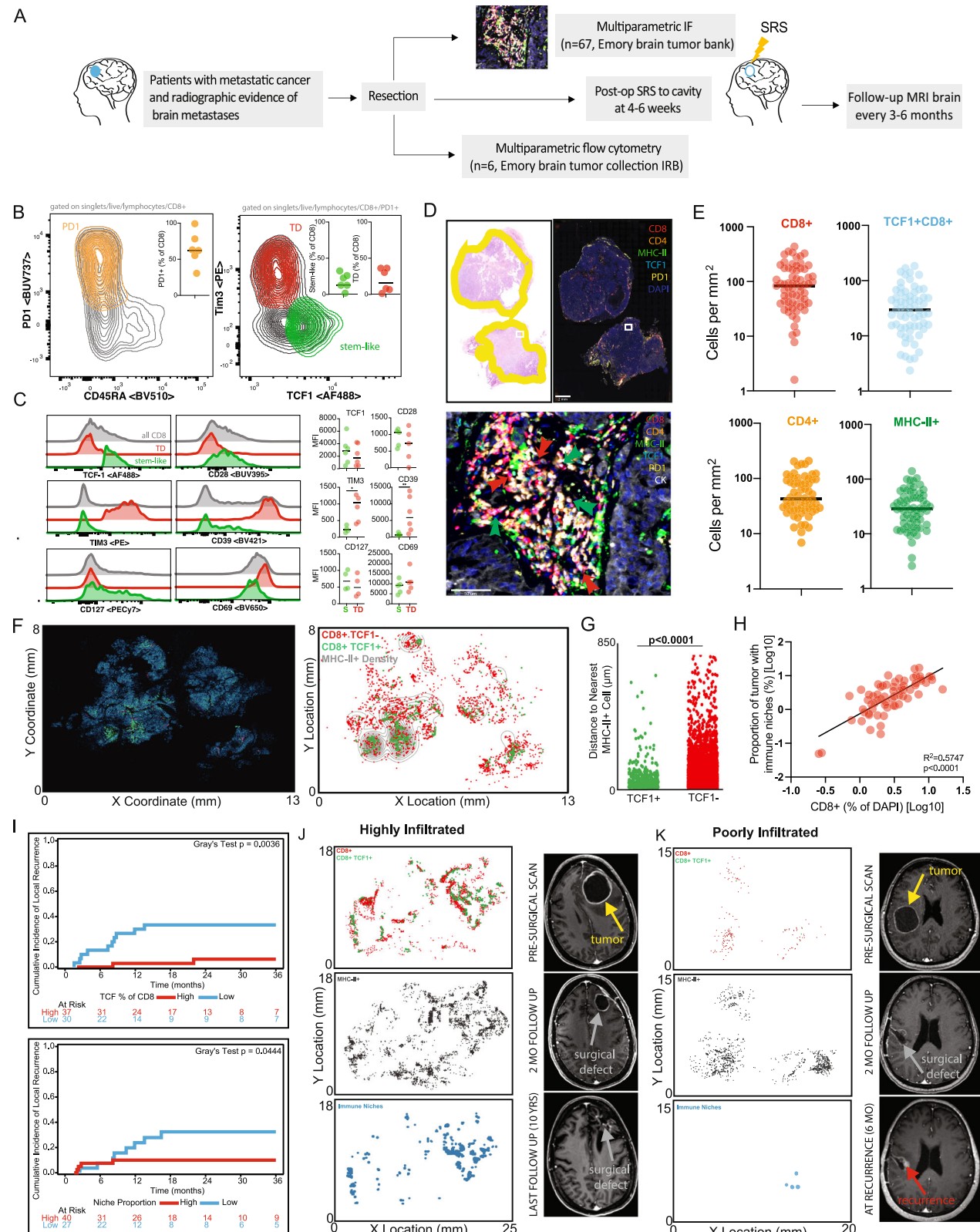

Supplementary Fig. 3A–D). Although total CD8 T cell numbers were reduced, the majority of infiltrating CD8 T cells in pSRS tumors exhibited a PD1+ phenotype similar to that of the Res cohort, with comparable proportions of stem-like (TCF1+) and effector-like (TCF1-) populations (Supplementary Fig. 3E, F). However, although similar in frequency, the numbers of both TCF1+ and TCF1- CD8 T cells were reduced in pSRS tumors (Fig. 3B). We validated these results by flow

cytometry, with a total reduction in CD8 T cells across pSRS BrM tumors (Supplementary Fig. 4A-B), while exhibiting similar PD1+ TCF1+ and TCF1- CD8 T cell phenotypes compared to Res BrM counterparts (Supplementary Fig. 4C-F). Interestingly, the number of MHC-II+ APCs were not significantly different between the two cohorts (Fig. 3B), indicating that pSRS may differentially impact immune populations in the brain. Given the maintenance of MHCII+ cells, we next assessed

**Fig. 2 | TCF1+ PD1+ stem-like T cells are found in brain metastases and reside in an immunological niche. A** Schema of sample collection, processing, and analysis of BrM tissue. **B** Flow cytometry characterizing PD1+ TCF1+ stem-like and PD1+ TIM3+ terminally differentiated (TD) effector-like cells in BrM. Greater than 50% of CD8 T cells in BrM are PD1+ . $n = 6$, with median shown. **C** Expression (mean fluorescence intensity, or MFI) of activation markers, checkpoint molecules, and transcription factors by TD and stem-like subsets, gated as in B. Statistical significance calculated by Mann-Whitney U test, with *:$p = 0.0411$ **:$p = 0.0043$. **D** H&E with tumor regions outlined in yellow (left); immunofluorescence whole slide image (right) with the region of interest highlighted by the white box and zoomed immunofluorescence image demonstrating immune cell cluster (below). **E** Quantitation of immune cell densities, $n = 67$, with the median shown. **F** Digital reconstruction of whole slide immunofluorescence (left) and cellular spatial relationship map (right) displaying a contour of MHC-II+ cellular density and the x, y locations of TCF1+/- CD8 T cells. **G** Distance between TCF1+/- CD8 T cells and the closest MHC-II+ cell. **H** Percentage of total CD8 T cells correlates with the proportion of tumor tissue occupied by immune niches ($n = 67$). **I** Cumulative incidence curves associating extended local control of disease with a higher percentage of TCF1+ of CD8 T cells (left, Gray's test two-sided, $p = 0.0036$) and with a higher tumor niche proportion (right, Gray's test two-sided, $p = 0.0444$). **J** x, y location maps of CD8 T cells, MHC-II+ cells, and immune niches in a highly infiltrated cystic BrM (left) with MRI images of the pre-operative and post-operative resection cavity followed for 10-years without evidence of local recurrence (right). **K** x, y location maps of a poorly infiltrated BrM (left). Nodular local recurrence at the surgical cavity margin occurred at 6 months post-operatively denoted by red arrow (right). Source data are provided as a Source Data file.

whether immune niche organization was impacted by pSRS. We found immune niche density in pSRS tumors was significantly reduced compared to Res tumors (Fig. 3C), while the positive correlation between CD8 T cell density and immune niches was maintained (Fig. 3D). Importantly, this relationship was not impacted by dexamethasone dose ($p = 0.990$) or BrM histologies (Supplementary Fig. 3G, K). In correlates from our prospective trial (Supplementary Fig. 5A–N) for the high dose (Arm B) and low dose (Arm A) of dexamethasone, we observed no significant differences in the density of total CD8 T cells or PD1+ CD8 T cell subsets per mm² between the study arms (Supplementary Fig. 6A–D). Nor was there a significant difference between MHC-II+ APC infiltration or immune niche density (Supplementary Fig. 6E–I). Overall, these data indicate that pSRS reduces local CD8 T cell numbers in BrM independent of dexamethasone dose, with limited effect on MHCII+ APCs. Importantly, PD1+ TCF1+ stem-like CD8 T cells and intratumoral immune niche organization were maintained.

We then assessed the temporal effects of pSRS on the immune infiltrates in our retrospective cohort. In patients who had BrM resected on the same day as pSRS (Day 0), there was no significant difference in the number of total CD8, TCF1+ , or TCF1- cells in the BrM compared to Res patients (Supplementary Fig. 7A–C). In contrast, tissue resected between 1-5 days post pSRS had significantly lower numbers per mm² of total CD8 T cells, as well as reduced TCF1+ , and a trend for lower TCF1- effector-like CD8 T cells (Fig. 3E–J, Supplementary Fig. 7D). Interestingly, this difference was not seen in BrM which were resected 6 or more days after pSRS. These patients exhibited a rebound of TCF1- effector-like CD8 T cells to baseline numbers, while PD1+ TCF1+ CD8 T cells remained depressed (Fig. 3I, J). Notably, these changes were also reflected in the frequency distribution of PD1+ CD8 T cell populations, with an increase in TCF1- CD8 T cell frequency and a concomitant reduction in the TCF1+ CD8 T cells (Fig. 3K, L). Although MHCII+ APC numbers did not significantly change over the treatment time course (Supplementary Fig. 7E, F), there was a trend for a reduced immune niche density that persisted through the 6+ timepoint (Fig. 3M, Supplementary Fig. 7G). This effect is consistent with the reduction in the TCF1+ CD8 T cell population,which primarily resides within these organized clusters. Overall, these data show a broad and rapid decrease in CD8 T cell numbers in BrM following pSRS. Although CD8 T cell numbers were reduced, BrM retained immune niche composition with MHC-II+ APCs following pSRS therapy. Additionally, our data suggest that TCF1- CD8 T cells rapidly rebound. Given the extensive data in pre-clinical models of TCF1+ CD8 T cells being the precursor to the TCF1- populations, we hypothesize that the relative abundance of TCF1- CD8 T cells at day 6+ following pSRS may be driven by differentiation of the TCF1+ population within the BrM.

**pSRS shifts the dendritic cell signature towards a pro-inflammatory phenotype and promotes the accumulation of the TD1 GZMB high effector subset.** To further examine the impact of pSRS on immune cells in BrMs, we performed single cell RNAseq on 13 pSRS BrM from the prospective trial and 18 Res BrM collected prospectively off trial. For this analysis, we separately sorted CD8 T cells and the remaining CD45+ cells from patients' tumors and mixed them 1:1 to enrich for infiltrating CD8 T cells (Supplementary Fig. 8A). Flow cytometry quantification showed a wide range of CD45+ cell infiltration, with pSRS patients exhibiting a lower frequency of total CD45+ cells in their tumors, which was consistent across pSRS dosage and fractionation (Supplementary Fig. 8B, C, Table 1). Transcriptional analysis identified six clusters of CD45+ immune populations within our BrM cohort of multiple histologies, including CD8 T cells *(CD3, CD8B)*, CD4 T cells *(CD3, CD4, FOXP3)*, activated B cells *(CD79A, MS4A1, IGHM)*, antibody secreting cells (ASCs; *MZB11, IGKC*), plasma cells *(MZB1, PRDM1, IGKC)*, and a grouped myeloid antigen presenting cell population (Macs/DCs; *CD68, ITGAX, ITGAM)* (Supplementary Fig. 8D–F). We found that pSRS BrM were enriched for activated B cells and consistently had a reduced frequency of CD4 T cells, indicating that pSRS may have differential effects on various immune populations (Supplementary Fig. 8G).

Given that immune niches are primarily composed of myeloid APCs, we next examined whether pSRS impacts their transcriptional program. After sub-clustering the myeloid APCs, we found four distinct populations, including macrophages (*ITGAM, CD68, FCGR3A, LYZ*), activated dendritic cells (*ITGAX, LYZ, IL1B, HSP1AB*), classical dendritic cells type 2 (cDC2s) (*ITGAX, SIRPA, CD109, CD300E)*, and monocytes (*ITGAM, CD14, C1QA)* (Supplementary Fig. 8H). We focused on activated DCs, given their consistent presence across patient BrMs, whereas the other APC population frequencies were highly variable and driven by single patient samples. Activated DCs were characterized by high expression of antigen processing machinery, co-stimulation markers, as well as cytokines and chemokines related to immune cell recruitment *(IL1B, CXCL8, CXCL16)* across both groups (Supplementary Fig. 8I). Interestingly, activated DCs from pSRS BrM had significantly higher expression of chemokines *CXCL3* and *CXCL2*, which are associated with inflammatory recruitment of innate immune cells, and higher expression of complement associated genes (Supplementary Fig. 8J). In agreement, activated DCs from pSRS enriched for pathways related to inflammatory immune response, recruitment of innate immune populations, and complement cascades (Supplementary Fig. 8K), suggesting pSRS may shift the myeloid populations towards a more inflammatory state.

Given that pSRS reduces local BrM CD8 T cell numbers (Fig. 4A), we next examined whether the transcriptional program of CD8 T cell populations was altered by pSRS. Single cell RNAseq analysis identified four populations of activated CD8 T cells, including stem-like, transitory, GZMB high effector (TD1), and GZMB low effector (TD2) cells, as well as a population of naive/memory CD8 T cells across both groups (Fig. 4B, C, Supplementary Fig. 9A, B). Stem-like CD8 T cells (cluster 1) were characterized by high expression of the transcription factor *TCF7*, encoding TCF1, and had low levels of effector and cytotoxic molecules (Fig. 4C and Supplementary Fig. 9B). These *TCF7*+ CD8 T cells transcriptionally differed from naive/memory CD8 T cell subsets, with low

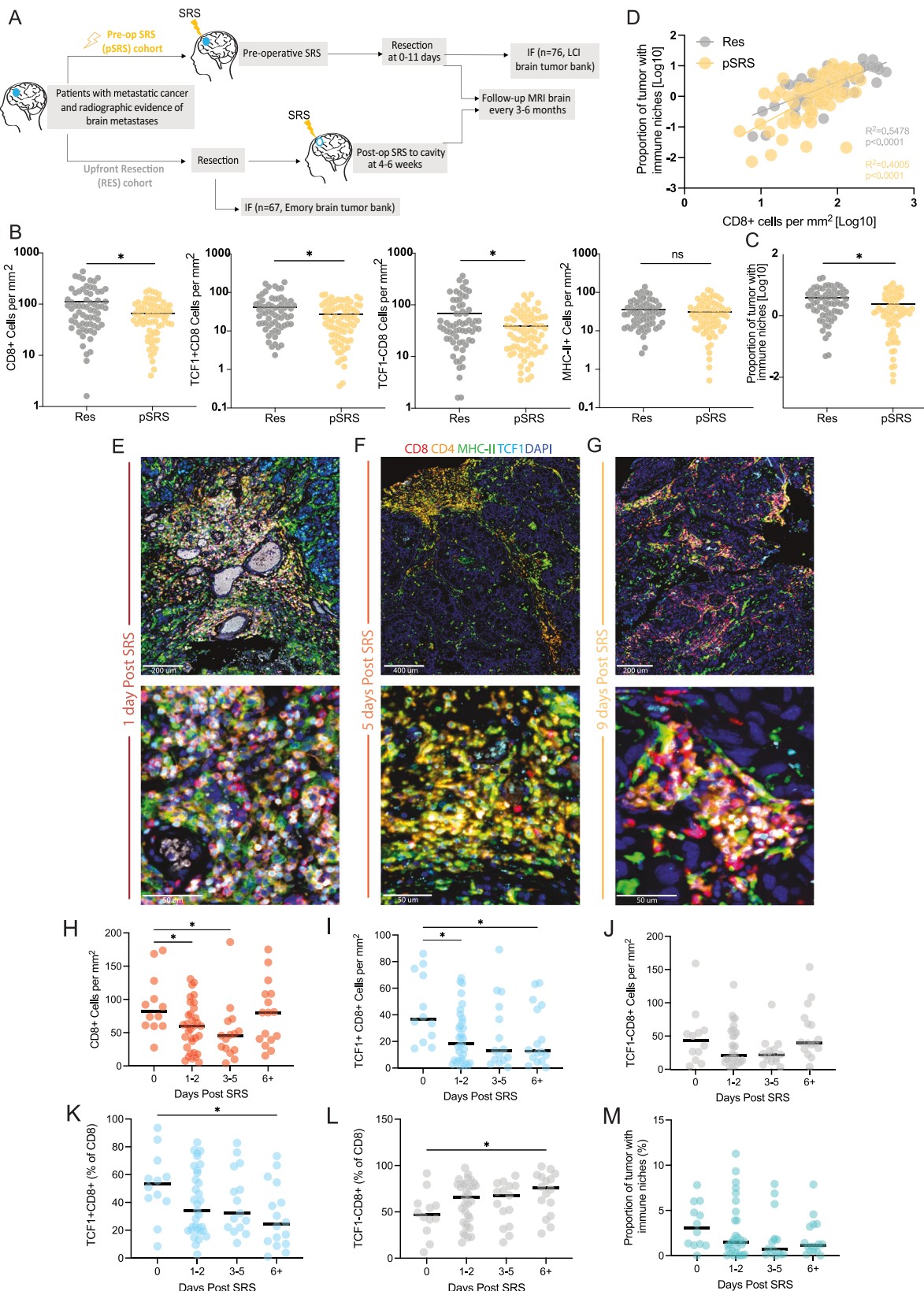

expression of *SELL* and increased expression of activation associated genes *CXCR3, GPR183* and LGALS3 (Fig. 4C). Cluster 1, *TCF7*+ CD8 T cells enriched for the human stem-like signature, previously defined in antigen specific CD8 T cells in HPV+ head and neck tumors[8] (Supplementary Fig. 9C), further supporting a precursor-like phenotype. Cluster 2 was defined as a transitory population, characterized by high

expression of *CD69*, the transcription factors *JUNB* and *FOS*, TCR signaling dependent gene *NR4A2*, and effector cytokine *IFNG* (Fig. 4C and Supplementary Fig. 9B). The remaining activated CD8 T cells had an effector transcriptional profile with low expression of *TCF7*, high levels of *PRDM1*, along with various cytotoxic molecules and inhibitory receptors (Supplementary Fig. 9B). While both cluster 3 and cluster 4

**Fig. 3 | The impact of pre-operative SRS on the immune niche. A** Schema of sample collection, processing, and analysis of BrM tissue including upfront resection (Res) and pre-operative SRS (pSRS) specimens. **B** Comparison of quantified immune cell infiltrate in Res and pSRS samples. Total CD8 cells per mm$^2$ are lower in pSRS samples compared to Res ($p$= 0.0160), TCF1+ are lower in pSRS samples (*:$p$= 0.0223), TCF1- are lower in pSRS (*:$p$= 0.0452), and MHC-II+ cell per mm$^2$ is similar in Res vs pSRS ($p$= 0.2642). **C** Proportion of tumor with immune niche is lower in pSRS compared to Res (*:$p$= 0.0022). In B & C, calculated by Mann–Whitney U test, $n$= 67 Res, $n$= 76 for pSRS. **D** Correlation between CD8 cells per mm$^2$ and proportion of tumor with immune niche is maintained in pSRS BrM ($R^2$ = 0.4005, p < 0.0001, as calculated by linear regression; $n$= 67 for Res, $n$= 76 for pSRS). **E–G** Representative immunofluorescence of the immune niches and T cell subsets **E** at day 1, **F** at day 5, **G** and at day 9 after pSRS. **H–M** Quantification of immune cell density at 0, 1-2, 3-5, or 6+ days following pSRS, statistical significance calculated by one-way ANOVA, $n$= 76. **H** Number of CD8 cells per mm$^2$ decreases from day 0 to day 5 and then rebounds to baseline by day 6+ following pSRS with *:$p$= 0.0424, *:$p$= 0.0285. **I** Number of TCF1+ CD8 cells per mm$^2$ decrease following pSRS and remain lower at day 6+ with *:$p$= 0.0198, *:$p$= 0.0447. **J)** Number of TCF1- CD8 cells per mm$^2$ have a numeric, but not statistically significant decrease from day 0-5 with a rebound by day 6+ . **K** Percentage of TCF1+ CD8 cells (of total CD8 T cells) significantly decreased from day 0 to day 6+ with *:$p$= 0.0341. **L** In contrast, the percentage of TCF1- CD8 cells (of total CD8 T cells) significantly increased from day 0 to day 6+ with *:$p$= 0.0341. **M** Proportion of tumor occupied by immune niches showed a numeric but not statistically significant decrease from day 0 to day 6 + . In H-M, $n$= 12, 33, 15, 16 for days post SRS 0, 1-2, 3-5 and 6+ respectively. IF (immunofluoresence). Source data are provided as a Source Data file.

populations enriched for an effector signature (Supplementary Fig. 9D), we detected a distinct pattern of expression across cytotoxic molecules, transcription factors, and inhibitory receptors. A subset of effector CD8 T cells (TD1) expressed high levels of *GZMB, PRF1, CX3CR1*, and *GNLY* while a second population (TD2) expressed *GZMK, EOMES, TIGIT*, and *LAG3* (Fig. 4C and Supplementary Fig. 9B), suggesting distinct differentiation states.

Notably, although both Res and pSRS CD8 T cells clustered together (Fig. 4D, E and Supplementary Fig. 9A), pSRS BrM exhibited a significant enrichment of CD8 T cells in cluster 3 (TD1 GZMB high) and a trend towards a lower proportion of cells within clusters 2 (Transitory) and 4 (TD2 GZMB low) (Fig. 4D). Evaluating these populations as a function of time from pSRS to surgery, we found a trend towards an increase in TD1 *GZMB+* CD8 T cells with a concomitant reduction in *TCF7+* stem-like CD8 T cells frequency at 6+ days after pSRS (Supplementary Fig. 9E). In the overall pSRS group, these CD8 T cell populations were not significantly altered by high relative to low dose dexamethasone (Supplementary Fig. 9F). Furthermore, these differences in frequency within cluster were no longer significant when comparing CD8 T cell distribution as a proportion of total CD45+ cells, due to the reduced overall CD8 T cell infiltration in pSRS BrM (Fig. 4E).

To gain deeper insight into the transcriptional changes induced by pSRS on CD8 T cell subsets, we performed differential gene expression (DEG) analysis across treatment groups. Stem-like, *GZMB* high TD1, *GZMB* low TD2, and transitory CD8 T cells exhibited 74, 135, 194, and 118 upregulated genes in pSRS BrMs, respectively (Supplementary Fig. 9G). The stem-like, TD2 and transitory clusters had enrichment in pathways related to apoptosis, DNA damage and cellular response to stress (Fig. 4F, Supplementary Fig. 9G-I). In contrast, *GZMB* high TD1 effector cells showed enrichment for pathways related to cytotoxicity and effector differentiation (Fig. 4G, Supplementary Fig. 9G). Among the top upregulated genes in this population were *KLF2* and natural killer (NK) receptors *KLRG1, KIR3DL1*, which have previously been associated with exhausted CD8 T cells and enhanced effector functions (Fig. 4H, Supplementary Fig. 9G). In contrast, CD8 T cells from Res BrMs showed enhanced expression of chemokine receptors *(CXCR3, CXCR6)*, cytokine signaling pathways *(STAT1)*, interferon induced genes *(IFITM1, OASL)*, and inhibitory receptors *(TIGIT)* (Fig. 4H, Supplementary Fig. 9G). Overall, these data indicate that pSRS induces a reduction in total CD8 T cell infiltration and promotes transcriptional changes associated with cellular stress, DNA damage repair, and cytotoxicity among PD1+ CD8 T cell subsets.

Given the co-localization of CD8 T cells near APC-rich immune niches, we next explored the possible interactions between these cells and how they might be modulated by pSRS. We performed Niche-Net sender-receiver analysis between the top differentially expressed ligands on activated DCs and differentially expressed receptors on CD8 T cell subsets from Res and pSRS groups. Res BrM DCs expressed high levels of lymphotoxin alpha (*LTA*) and other TNF family members, which were predicted to interact with TNF receptors across CD8 T cell populations (Fig. 4I). Moreover, DC *CXCL9* and *CCL2* were predicted as top chemokine receptor pairs to recruit *CXCR3* and *CCR5* expressing CD8 T cell subsets, which might facilitate co-localization of these populations in immune niches in the BrM (Fig. 4I). Notably, pSRS altered the array of ligand-receptor pairs expressed among these populations. Chemokine receptor CXCR3 was no longer among the top predicted interactions, partially due to reduced expression across CD8 T cell populations in pSRS BrM (Fig. 4J). Instead, TNF family members related to apoptotic pathways and intrinsic inhibitory signals, including *FASLG, TNFSF10* (TRAIL), and *TNFSF14* were among the top predicted interactions (Fig. 4J). Moreover, this increased expression of ligands related to TNF family members in pSRS DCs was accompanied by an increased expression of lipid transporter apolipoprotein E (*APOE*) (Fig. 4J). *APOE4* is associated with neurodegenerative tauopathy, and a recent study demonstrated that overexpressing human *APOE4* intracranially in mice lead to significant increases in neuroinflammation and cytotoxic T cell[31]. Together, these data indicate that pSRS shifts the transcriptional signatures of dendritic cells towards both a pro-inflammatory and an apoptosis-inducing phenotype. These activated DCs may deliver these signals to PD1+ CD8 T cells that promote both the accumulation of the TD1 cytotoxic subset and cell death, contributing to the overall reduction in the CD8 T cell population across BrM tumors.

**Immunodominant clonotypic overlap is observed between the CD8 T cell subsets in Res and pSRS BrM.** Our immunofluorescence, flow cytometry, and transcriptional data suggest that PD1+ TCF1+ CD8 T cells can act as tumor-reactive stem-like precursors to the TD1 and TD2 effector populations in BrM. To gain further insight into the relationship between these CD8 T cell subsets, we performed matched VDJ sequencing analysis from the single cell RNAseq above on pSRS samples from our prospective trial or Res samples collected prospectively off trial. We first assessed the clonal expansion of each CD8 T cell subset across groups. Consistent with their respective phenotypes, PD1+ TCF1+ stem-like, transitory, and TD effector populations shared immunodominant and subdominant expanded clones in both Res and pSRS BrM (Fig. 5A, B and Supplementary Fig. 10A, B), indicative of lineage relationship and tumor reactivity. Additionally, all the activated stem-like, transitory, TD1, and TD2 CD8 T cell populations had a lower diversity TCR score when compared to the naive/memory cluster (Supplementary Fig. 10C). Furthermore, there were fewer unique clones along the effector differentiation pathway (stem-like to effector) with a high degree of clonal overlap between stem-like and TD1/TD2 in both Res and pSRS (Supplementary Fig. 10D, Fig. 5C). When accounting for the number of unique clones and clone size in the stem-like T cell repertoire, there was around 40% TCR clonal overlap with the TD1 population and 50% overlap with the TD2 population in both groups (Fig. 5C). In agreement, stem-like and TD populations shared a high degree of TCR similarity as measured by the Morisita-Horn index (Supplementary Fig. 10E), further indicating the *TCF7* expressing stem-like CD8 T cell population acts as a precursor to TD1 and TD2 CD8 T cells within BrM.

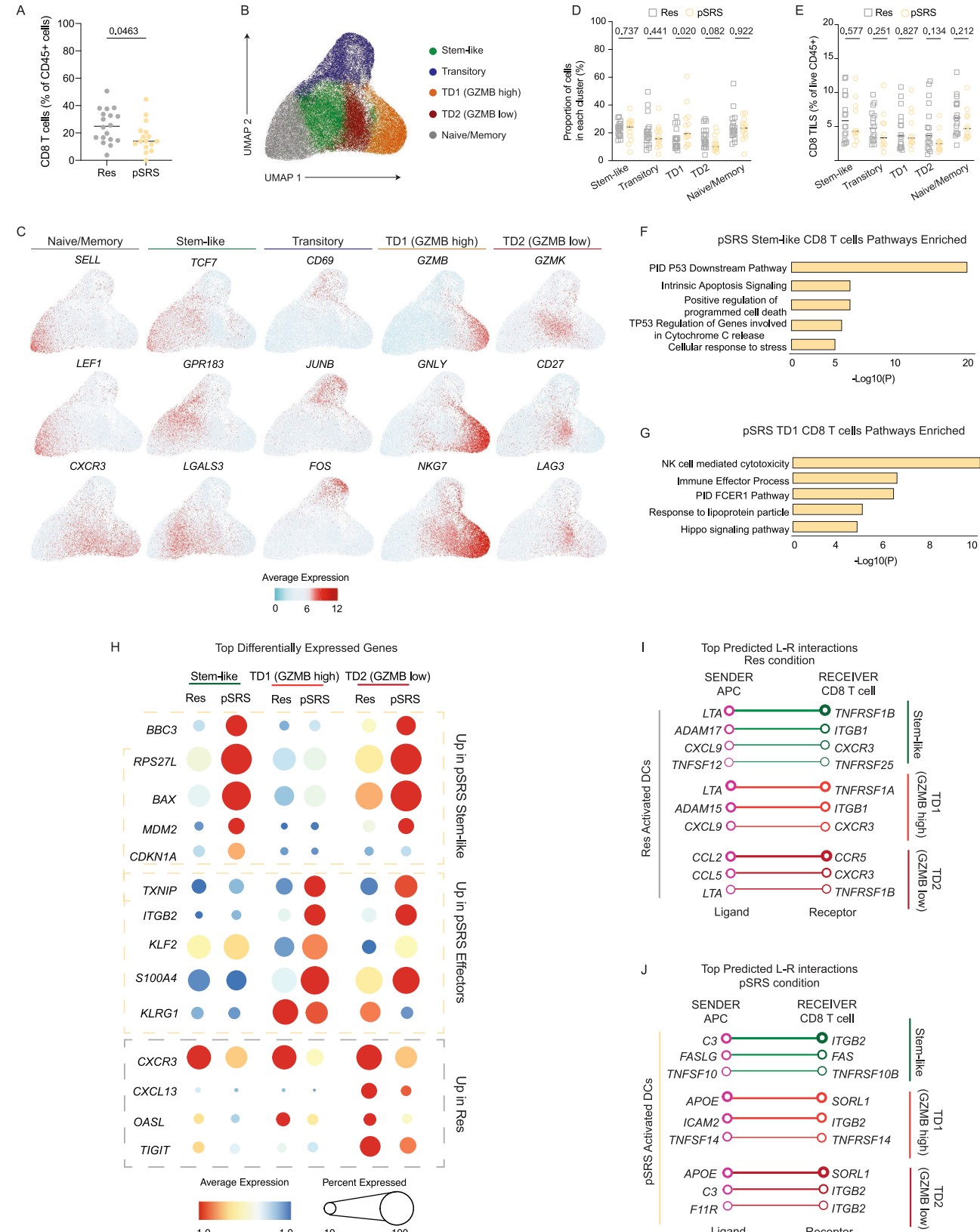

TD1 and TD2 populations also had a very high degree of TCR similarity (Supplementary Fig. 10E) suggesting that TD2 may reflect a more exhausted differentiation state of TD1. Phenotypically this was also apparent, given the higher expression of exhaustion markers (*TOX, PDCD1, LAG3*) and reduced expression of *GZMB* (Fig. 4C, Supplementary Fig. 9B). Interestingly, there was a trend for a greater

proportion of dominant TCR clones in the TD1 population in the pSRS group when compared to Res (Fig. 5D). This may indicate that pSRS boosts differentiation to the TD1 *GZMB* high effector population or preferentially affects subdominant clones. Together, these data support that *TCF7*+ stem-like CD8 T cells are the precursors to effector CD8 T cells within BrMs. Importantly, the lineage of differentiation

**Fig. 4 | pSRS shifts the dendritic cell signature towards a pro-inflammatory phenotype and promotes the accumulation of the TD1 GZMB high effector subset. A** Frequency of CD8 T cells of total live CD45+ cells across all patient tumors, as measured by flow cytometry. Medians are shown, and statistical comparisons were performed using two-sided unpaired Mann Whitney U test (*n*= 34). **B** Single cell RNAseq UMAP projection of sub-clustered CD8 T cells. **C** Normalized gene expression of selected differentially expressed genes defining each CD8 T cell cluster. **D**, **E** Relative cluster distribution **D** and total frequency of CD45+ live cells **E** of CD8 T cells subsets in Res (*n*= 18) and pSRS (*n*= 13) patient tumors. Medians are shown for each summary plot and statistical comparisons were performed using Mann Whitney U test between Res and pSRS groups. **F**, **G** Metascape pathway

enrichment analysis in CD8 T cell subsets from pSRS tumors. Top five significant pathways are shown for **F** Stem-like CD8 T cells and **G** TD1 GZMB high CD8 T cells. **H** Transcriptional comparison of CD8 T cell subsets across Res and pSRS groups. The color and size of the circles represent the normalized expression and proportion of cells expressing that gene, respectively. **I**, **J** NicheNet sender-receiver analysis showing the top 3-4 predicted ligand-receptor interactions between activated DCs and CD8 T cell subsets in **I** Res or **J** pSRS. The thickness of circles and connections represents the prioritization score and interaction strength for each L-R pair. *P* values are shown. Source data are provided as a Source Data file and are available on the NCBI Gene Expression Omnibus (GEO) database.

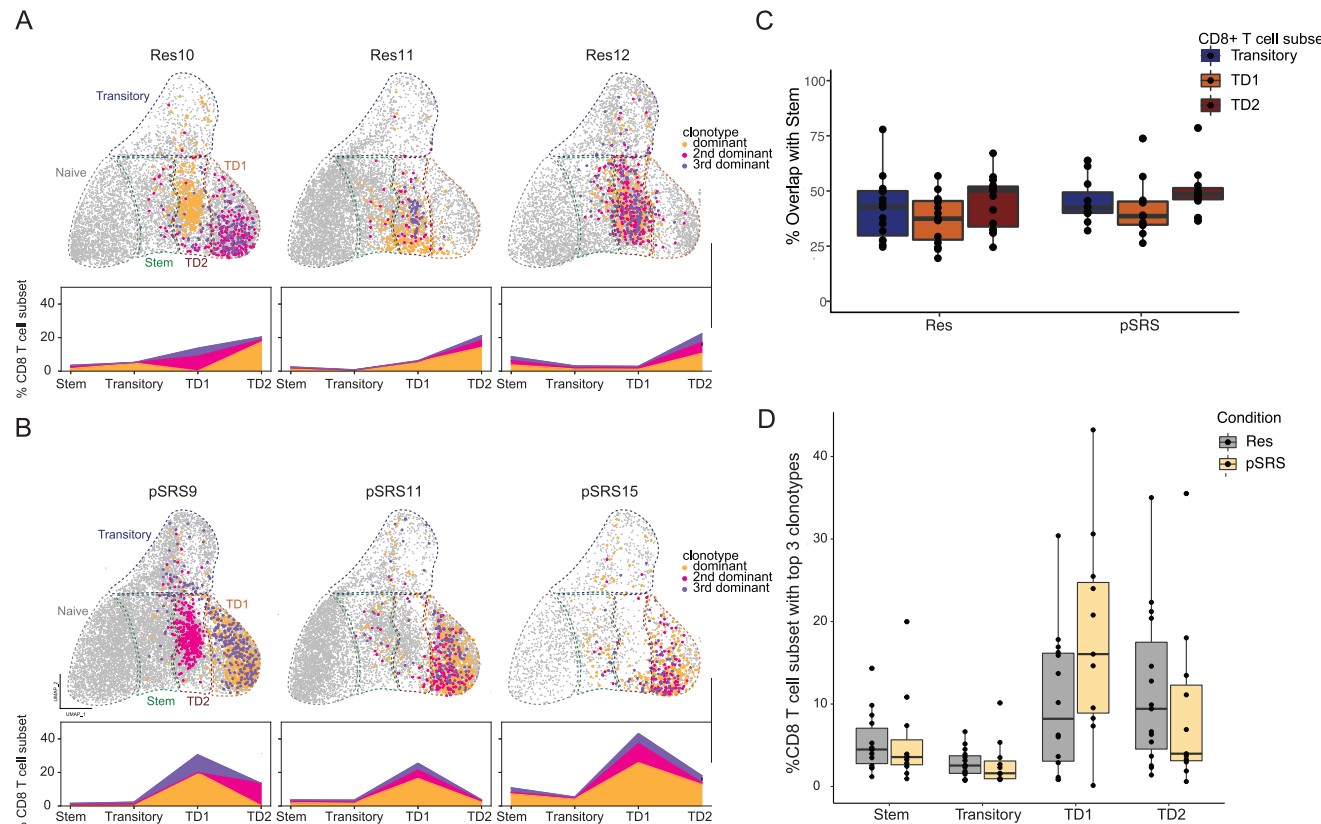

**Fig. 5 | Immunodominant clonotypic overlap is observed between the CD8 T cell subsets in Res and pSRS BrM.** Top 3 clonotypes for a representative sample of individual **A** Res or **B** pSRS BrM superimposed on the UMAP of the previously described CD8 T cell subsets (stem-like, transitory, TD1, and TD2), percentage of all cells quantified per BrM below. Highlighted clonotypes are present in all four subsets. **C** Percentage TCR overlap of stem-like CD8 T cells with Transitory, TD1 or TD2 in Res (left, *n*= 15) and pSRS (right, *n*= 11). Box plot shows min, max, median and

interquartile range. **D** Quantification of the top 3 clonotypes for all BrM grouped by cluster and subdivided by treatment. Immunodominant clonotypes were enriched for in the effector subsets compared to the stem-like and transitory subsets (*n*= 15 Res, *n*= 11 pSRS). Box plot shows min, max, median and interquartile range. Source data are provided as a Source Data file and are available on the NCBI Gene Expression Omnibus (GEO) database.

progressing from the TCF1+ stem-like to effector clusters remained intact following pSRS treatment.

## Discussion

In this study, we sought to understand the clinical and immunologic impact of pSRS for BrM. We found that pSRS could be delivered safely with clinical outcomes similar to BrM receiving post-operative SRS irrespective of dexamethasone dose (Fig. 1). Using multiple data sets including samples from our prospective trial, we found that immune niches comprised of TCF1+ CD8 T cells and antigen presenting cells were present at varying densities in Res and pSRS BrM. Importantly, these immune niches were associated with longer local control of disease (Fig. 2)[7], which were not affected by peri-operative dexamethasone dosage (Fig. 1, Supplementary Fig. 6). In the pSRS cohort,

while there was a transient reduction in BrM total and CD8 T cell subsets relative to Res, CD8 T cell populations appeared to rebound by day 6 post pSRS, driven by an increase in the frequency of effector-like TCF1- CD8 T cells (Fig. 3). Single cell RNAseq and VDJseq analysis found robust TCR clonal overlap between TCF1+ and effector CD8 T cell subsets across both Res and pSRS patients, indicating these cells have stem-like properties and may serve as the precursors to effector CD8 T cell populations within BrM. Finally, transcriptional analysis showed increased expression of inflammatory and apoptotic genes in pSRS dendritic cells and CD8 T cell subsets that could lead to an accumulation of GZMB high effector CD8 T cells within BrM (Figs. 4, 5).

The clinical trial had a few limitations. First, long term safety evaluation are needed to draw more extensive clinical conclusions. Second, it was not powered for secondary outcomes including

comparing the low and high dose steroid Arms. Therefore, although no significant differences in outcomes were observed, validation on a larger trial will be required.

Taken together, these immunological data fit into and build upon an expanding body of work evaluating the tumor-microenvironment in BrM[19,26,32–34]. Prior investigation of the spatial immune landscape of BrM has demonstrated a wide variety of immune clusters, some of which appear to confer prognostic significance[20]. One particularly elegant study compared the microenvironment of melanoma BrM and extracranial metastases and found that the BrM had higher frequencies of monocyte-derived macrophages and of exhausted TOX + CD8 T cells[19]. With our study, we did not seek to verify these comparisons, but rather to deeply interrogate the BrM CD8 T cell populations under both baseline (Res) and pSRS conditions. Our study is hypothesis-generating but does have a few limitations. Due to the sample size, a multivariate analysis for immune niche density and local control was not possible. Additionally, we did not find that measuring bulk CD8 T cells had prognostic value, which may be due to a less robust association of CD8 T cells alone with outcomes, leading to underpowering. Importantly, we selected local control as an end point as it has fewer confounders than distant brain failure (which is impacted by systemic disease burden) or overall survival (which has innumerable confounders). Validation of the immune niche in an independent BrM cohort, as well as further expanding the number and diversity of histologies of the BrM evaluated is the focus of our ongoing studies and will help determine whether it is a clinically useful biomarker. Future investigation of other endpoints beyond local control will also be important, as BrM CD8 T cell infiltration has been shown to be associated with improved survival[35]. Given the focus of this study was the CD8 T cell response, a deeper analysis of APC subtypes in the immune niche also requires further investigation. Here, by scRNA seq, we confirmed there are macrophages, dendritic cells, and B cells in the BrM, as also shown by other investigators[19,32] and the transcriptional profile of dendritic cells is impacted by pSRS towards an inflammatory state.

The pSRS findings were unexpected. We had hypothesized, based on our preclinical studies using non-CNS tumors, that pSRS would increase the density of stem-like CD8 T cells and the presence of immune niches around day 7 following radiation[25]. However, in our time from pSRS to surgery analysis in Fig. 3, we did not find an increase above baseline by day 6 + , but we did find a rebound in total CD8 T cell numbers that appeared to be driven by an increase in the frequency of the TCF1- subsets, particularly the TD1 GZMB + CD8 T cells. These results suggest that SRS, like anti-PD1 therapy, may drive both the proliferation and differentiation of stem-like CD8 T cells into more terminally differentiated, effector-like cells[5,36]. This mechanism would account for the rebound in total CD8 T cells and the relative changes in the frequency of these populations. The in-situ expansion and differentiation model is supported by the sustained clonotypic overlap between stem and terminal effector cells observed in the pSRS group. Alternatively, although less likely given these clonotypic data, SRS may promote the infiltration of terminal effector cells which differentiate from stem-like T cells at sites outside the CNS, including draining lymph nodes[12,25,37]. The differences between these human results and our prior pre-clinical work suggest that pre-clinical models may not fully recapitulate the human CNS immune response following SRS[25]. Further studies of in vivo and in vitro systems are necessary to interrogate these different possibilities and develop a more complete understanding of the similarities and differences between model systems and human patients.

Radiation has been shown to also induce dendritic cell maturation via the release of endogenous adjuvants (i.e., damage associated molecular patterns, or DAMPs)[38,39]. This stimulatory effect may explain the enhanced inflammatory response in DCs as well an immune effector process in the TD1 cells by single cell RNA seq in the pSRS

group. The immunogenicity of radiation is well known to depend on both the total dose and number of fractions[40]. In the 76 patients receiving pSRS analyzed by immunofluorescence, 97% received a single fraction with a median and mode of 15 Gy. This relatively homogeneous data set allows for a more uniform analysis of the impact of SRS on T cells subsets, but it also makes it more difficult to draw conclusions regarding the immunogenicity of other dose/fraction regimens in the CNS. This was investigated in our smaller prospective clinical trial dataset without any statistically significant results. This is a critically important question and will be the focus of future studies. Additionally, glucocorticoids, including dexamethasone, are known to impact T cell survival and CD8 T cell differentiation[27,28]. Here, we did not find a significant effect of dexamethasone dose on BrM T cell numbers in either the 67-patient retrospective Res cohort or in the clinical trial, which is consistent with other studies[41]. Importantly, despite this high dexamethasone dose, there was still significant BrM infiltration detected, demonstrating that dexamethasone does not eliminate nor appear to substantially impair the intracranial T cell response at the time points evaluated.

Broadly, these findings confirm that, despite the brain having several barriers to T cell infiltration, an immune response is capable of being mounted that resembles that of many other tumor types/locations and benefits patient outcomes. Clinically, these data suggest that the immune niche may not only be an important prognostic factor for outcomes, but also be predictive of an intracranial response to immunotherapy given the known importance of both stem-like CD8 T cells and co-stimulation for a robust response to anti-PD1 therapy[5,36,42]. These results also provide some insight into the optimal timing of surgical resection or immunotherapy strategies following SRS, while complementing the existing compelling data that combination SRS+immunotherapy may improve clinical outcomes[43,44]. Our data suggest that resection of BrM <6 days following pre-operative SRS limits any immunostimulatory benefit of SRS, due to the acute decline in CD8 T cells. Furthermore, anti-PD1 administration at a time point where CD8 T cell numbers are limited likely also blunts potential synergy of combinatory SRS and anti-PD1 therapy. These data can, therefore, inform investigation into optimal sequencing and combination of multiple therapeutic modalities. Future clinical studies are needed to further reveal the immunostimulatory potential of SRS, validate the prognostic value of immune niches, and determine whether an intra-cranial abscopal response can be generated.

## Methods

The research complies with all relevant ethical regulations. Both the prospective trial and the retrospective tissue analysis were approved by the Emory University Institutional Review Board. Informed consent was obtained for all patients enrolled on the prospective trial and prior to banking of tissue for those in the retrospective cohort.

### Sources of samples

The source of all samples are detailed in the respective figures/figure legends as well as Supplementary Table 6. Broadly, there are two retrospective cohorts: upfront resection (Res) and pre-operative SRS (pSRS) used for IF, and two prospective cohorts (also Res and pSRS) used for the remaining analyzes. The pSRS prospective cohort samples were all collected on the Emory University Winship Cancer Institute institutional pre-operative SRS clinical trial (NCT04895592). The tissue originated from two institutions (Emory University and the Levine Cancer Institute (LCI)) and are as follows: (1) Upfront resection (Res) immunofluorescence samples in Figs. 2, 3 are from the Emory University brain tumor bank (Supplementary Table 2); (2) pre-operative SRS (pSRS) immunofluorescence samples used in Fig. 3 are from the LCI brain tumor bank (Supplementary Table 4); (3) Res flow cytometry and scRNA seq samples in Figs. 2, 4, 5 were collected at Emory University Winship Cancer Institute utilizing our tissue collection protocol

IRB #00045732; (4) pSRS immunofluorescence, flow and scRNA seq samples in Figs. 4, 5 were collected on pre-operative SRS clinical trial (NCT04895592) (Supplementary Table 1).

## Emory University pre-operative SRS clinical trial

NCT04895592 is a two-arm non-randomized pilot study allocating patients with BrM to low dose Arm A (≤ 4 mg dexamethasone) or high dose Arm B (≥ 16 mg dexamethasone) in an alternating fashion. Both arms receive pSRS followed by resection. This trial accrued patient exclusively at Emory University Winship Cancer Institute. Eligible patients were ≥ 18 years of age, had a prior pathologically confirmed or suspected extracranial diagnosis of malignancy, brain metastases were visible on MRI, ECOG ≤ 2, a life expectancy >12 weeks and willingness and ability to tolerate both pre-operative SRS and surgical resection. Primary safety endpoint: <33% of patients develop grade ≥ 3 adverse event at 4 months. The sample size was determined based off a true toxicity event rate of 10%. This study design and conduct complied with all relevant regulations regarding the use of human study participants and was conducted in accordance with the criteria set by the Declaration of Helsinki. Additional details are available on clinicaltrials.gov. Time zero for all clinical outcomes was the date of pre-operative SRS delivery. First patient was enrolled on 8/11/2021, last treated patient enrolled on 4/21/2023. The protocol was amended on 4/11/23 to allow for a minimum of 10 evaluable patients per arm.

## Patient outcomes

Records of patients treated at two institutions (Emory University and the LCI) between 2007-2021 were evaluated and reviewed. Data were de-identified according to the Health Insurance Portability and Accountability Act, and all investigations were performed in accordance with the relevant guidelines and regulations. Patient characteristics of each cohort are shown in Supplementary Tables 2 and 4. Emory University comprised the Res cohort, and LCI comprised the pSRS cohort.

All clinical outcomes data are derived from the immunofluorescence samples from Emory University brain tumor bank as these patients had not received pSRS and had long-term follow-up available. Patient samples were selected which had a pathologic diagnosis of metastatic cancer to the brain and no prior immunotherapy. Immune biomarkers evaluated by patient characteristics are shown in Supplementary Table 3.

## Defining local recurrence

Local recurrence was defined based on the development of a contrast-enhancing mass within or adjacent to the prior resection cavity on MRI. If a re-resection was performed, pathologic confirmation was also utilized, although not required. If there was a question of the enhancement representing local recurrence vs. radiation necrosis, additional advanced imaging (e.g., MR perfusion, MR spectroscopy, or brain positron emission tomography) was obtained, and consensus was reached in a multidisciplinary neuro-oncology tumor board. Additionally, the lesion was followed over time, and persistence or resolution with observation or glucocorticoids further assisted with the differentiation of local recurrence vs. radiation necrosis.

## FFPE samples

Formalin fixed paraffin embedded tissue samples from these patients were stained and analyzed. 67 standard of care samples were obtained from the Emory Brain Tumor Bank, and 76 pre-operative SRS samples were acquired from LCI.

## FFPE sample preparation

Sections were deparaffinized in successive incubations with xylene and decreasing concentrations (100, 95, 75, 50, 0%) of ethanol. Antigen retrieval was achieved using Abcam 100x TrisEDTA Antigen Retrieval Buffer (pH = 9) heated under high pressure. Sections were then washed in PBS + 0.1% Tween20 before antibody staining.

## Immunofluorescence staining

Immunofluorescence antibody staining was done using two different techniques: (1) Sections were blocked for 30 min with 10% goat serum in 1x PBS + 0.1% Tween20. Sections were then stained with appropriate primary and secondary antibodies. Primary antibodies were incubated for 1 h at room temperature. Secondary antibodies were incubated for 30 min at room temperature. Detailed information about antibodies used are listed in Supplementary Table 7. Sections were counter-stained with DAPI according to manufacturer instructions (Thermo-Fisher). (2) Immunofluorescence antibody staining was performed using the Opal 7-color immunofluorescence kit (Akoya Biosciences) endogenous peroxidase activity was quenched by microwave treatment of the slides with AR buffer. Non-specific binding was blocked with blocking/antibody diluent. After incubation with the primary antibody, the slides were incubated with HRP Mouse+Rabbit secondary antibody and then incubated in the appropriate opal fluorophore for 10 min. The slides were counterstained with DAPI.

## Image capture and analysis

The selected fluorophore panel (1) allowed for simultaneous visualization of three targets and a nuclear stain (DAPI) using a Zeiss Axio Scan.Z1 Slide Scanner equipped with a Colibri 7 Flexible Light Source. Zeiss ZenBlue software was used for post-acquisition image processing. Slides stained with the Opal IHC Kit (2) were scanned using a Perkin Elmer Vectra Polaris and allowed for simultaneous visualization of six targets and a nuclear stain. For brightfield imaging, slides were scanned using a Hamamatsu's Nanozoomer slide scanner. Images were analysed using CellProfiler, QuPath, and custom R and python scripts, as previously described[7]. This analysis pipeline allowed for determination of the x, y location of each cell in each image, as well as the quantitation of the distance between cells and the density of each cell type. Immune niches were defined as 100um x 100um cellular neighborhoods where both TCF1+ CD8 T cells and MHC-II+ antigen presenting cells co-localized. Proportion of tumor with immune niches was defined as the percentage of tumor tissue (percentage of 10,000um$^2$ neighborhoods) occupied by immune niches (where TCF1+ CD8 T cells and MHC-II+ cells co-localize in a local 10,000um$^2$ cellular neighborhood).

## Fresh human sample collection, processing and flow staining

BrM samples were collected after patients underwent craniotomy and surgical BrM resection. For the pSRS group, SRS was administered prior to craniotomy on our institutional clinical trial, NCT04895592. All fresh samples (Res and pSRS) for flow analysis were treated and acquired at Winship Cancer Institute. Samples were collected directly after resection into Phosphate Buffered Saline. The samples were cut into small pieces, digested with a MACS enzyme cocktail, and homogenized using a MACS Dissociator. The digested tumor was washed then through a 70um filter to obtain a single cell suspension. Samples were then preserved in freezing media (FBS + 10% DMSO) at −80°C.

Single cell suspensions from processed human tumor samples were stained with antibodies listed Supplementary Table 7. Live/dead staining was done using fixable near-IR or aqua dead cell staining kit (Invitrogen). Cells were permeabilized using the FOXP3 Fixation/Permeabilization kit (eBioscience) for 45 min at 4°C and stained with intracellular antibodies in permeabilization buffer for 30 mins at 4°C. Samples were acquired on a BD FACSymphony instrument and analyzed using FlowJo (v10).

## ScRNA/VDJ seq

Single cell suspensions were stained and sorted on the Beckton Dickinson FACS Aria II Cell Sorter to acquire CD45+ CD8- and CD45 + CD8+ populations. These two sorted populations were then mixed 1:1 with

the goal of enriching for the infiltrating CD8 T cell population. Single cell RNAseq libraries were made using the Chromium single cell 5′ Library and Gel Bead Kit (10x Genomics) and captured into the Gel Beads-in-emulsion (GEMs). $N = 34$ BrM were submitted for sorting, and sufficient cells were captured for $n = 31$ ($n = 18$ Res, $n = 13$ pSRS). After the reverse transcription GEMs were disrupted and cDNA was isolated and pooled. The barcoded cDNA was fragmented, and end repair and A-tailing was done, followed by sample index PCR. The purified libraries were sequenced to 50,000 reads/cell on a HisSeq300 (Illumina) with 26 cycles for read 1, 8 cycles for index (i7) and 91 cycles for read 2. Paired VDJ sequencing was also performed. Filtered contig annotation files were compiled for each sample, and raw consensus ids were used to delineate clonotype. 72 cells with multiple clonotypes were removed. Samples with >50% of cells with missing TCF1 information were removed (Res1, pSRS4, pSRS13). Samples with dominant clonotype present in less than 20 cells were removed (Res1, pSRS13, Res5, pSRS7) leaving $n = 15$ for Res and $n = 11$ for pSRS.

Cellranger v3.1 was used to align, filter, and count the barcodes and unique molecular identifiers (UMI). Data was then analyzed using Seurat v3.0. Briefly, cells with less than 7% mitochondrial genes were used. Cells that expressed less than 500 genes or more than 6000 were excluded from analysis. Samples with at least 3 cells and 100 features were analyzed. Raw counts were then normalized for each UMI based on total expression, scaled by multiplying by 10,000, and then log transformed. Variable genes were determined based on average expression and dispersion, then used to perform principal component (PCA) analysis. Selected PCAs were used to generate clusters and UMAP plots. Heatmaps were generated using scaled expression data of marker genes, obtained using the FindAllMarkers function in Seurat. Normalized gene expression data was also shown as feature plots. Gene set scoring was performed using VISION R package V2.1. For clonotype analysis, MiXCR was run on raw fasq for Res and pSRS. The clonotypes identified were overlaid on the UMAP of the T cell subsets. The proportion of cells with top 3 dominant clonotypes were calculated for each CD8 T cell subset. The number of unique clonotypes were calculated for each CD8 T cell subset.

### Statistical analysis

The optimal cutoff values for stem-like T cells and immune niche proportion was determined by bias-adjusted log-rank test[45]. Local recurrence and death were regarded as two competing events. Cumulative incidence plots were created based on the proportional subdistribution hazards model, and Gray's test was performed to analyze the differences between high and low groups. SAS software version 9.4 (SAS Institute, Inc. Cary, NC) was utilized for the data analyzes.

### Reporting summary

Further information on research design is available in the Nature Portfolio Reporting Summary linked to this article.

## Data availability

Study protocol (for NCT04895592) is available in the Supplementary Information file. Additional de-identified participant data including dosimetric parameters will be made available upon request. Appropriate inquiries should be directed to the corresponding authors. scRNA-seq data are available in the NCBI Gene Expression Omnibus (GEO) database under the accession number GSE275929. The remaining data are available within the Article, Supplementary Information or Source Data file. Source data are provided in this paper. Source data are provided with this paper.

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

## Acknowledgements
This work was primarily supported by the American Cancer Society grant CSDG-22-049-01-IBCD to ZSB. Additional support came from National Institute of Health grants 1-K12-CA-237806-01 and 1-K08-CA-270401-01A1 to ZSB as well as National Cancer Institute grant 1-F30-CA-243250 for CSJ, National Cancer Institute grant 1-R01-CA280069 to HK, ASTRO-MRA #816282 for ZSB, DOD grant W81XWH-20-1-0525 to HK and VM, and Cancer Research Institute funding to HK. We would like to acknowledge the Yerkes NHP Genomics Core (P51 OD011132, NIH S10 OD026799) and the Emory Flow Cytometry Core (UL1TR002378).

## Author contributions
C.J., H.K., R.S.P., Z.S.B., and M.K.K. conceived of the study. P.C., B.T.V., C.Z., S.W., and R.L. acquired the immunofluorescence, single cell sequencing and flow cytometry data. P.C., C.S.J., and L.D.B. analyzed the immunofluorescence. B.H., T.D., M.A., I.G., and V.D. acquired the patient data. S.G.N., S.L. provided annotation of the pathology slides. K.B.H., J.Z., J.J.O., E.K.N., K.A.H., and P.P. enrolled patients on study and assisted with tissue acquisition. R.S.P., K.P., S.H.B., and A.L.A. provided tissue and assisted with pSRS retrospective dataset. S.G. and J.M.S. performed all clinical statistical analyzes. M.S.P., N.T.P., C.Y., M.A.C., H.C., N.P., and A.K. analyzed and interpreted the scRNA seq data. All authors contributed to writing and editing the manuscript including M.A.T., A.G.S., and A.H.K.

## Competing interests
The authors declare no competing interests.

## Additional information

Caroline S. Jansen [1,18], Meghana S. Pagadala [2,18], Maria A. Cardenas [1,18], Roshan S. Prabhu[3], Subir Goyal[4], Chengjing Zhou[5,6], Prasanthi Chappa[5,6], BaoHan T. Vo[1], Chengyu Ye[1], Benjamin Hopkins[5,6], Jim Zhong[5,6], Adam Klie[7], Taylor Daniels[5,6], Maedot Admassu[5,6], India Green[5,6], Neil T. Pfister[8], Stewart G. Neill[9], Jeffrey M. Switchenko[4], Nataliya Prokhnevska [1], Kimberly B. Hoang[6,10], Mylin A. Torres[5,6], Suzanna Logan[11], Jeffrey J. Olson[6,10], Edjah K. Nduom[6,10], Luke del Balzo[1], Kirtesh Patel[12], Stuart H. Burri[3], Anthony L. Asher[13], Scott Wilkinson [14], Ross Lake[15], Aparna H. Kesarwala [5,6], Kristin A. Higgins[5,6], Pretesh Patel[5,6], Vishal Dhere [5,6], Adam G. Sowalsky [14], Hannah Carter [16], Mohammad K. Khan[5,6], Haydn Kissick [1,17,19] ✉ & Zachary S. Buchwald [5,6,19] ✉

[1]Department of Urology, Emory University, Atlanta, USA. [2]Department of Medicine, Memorial Sloan Kettering Cancer Center, New York, USA. [3]Southeast Radiation Oncology Group, Levine Cancer Institute, Atrium Health, Charlotte, USA. [4]Department of Biostatistics and Bioinformatics, Rollins School of Public Health, Emory University, Atlanta, USA. [5]Department of Radiation Oncology, Emory University, Atlanta, USA. [6]Winship Cancer Institute, Emory University, Atlanta, USA. [7]Biomedical Sciences Program, University of California San Diego, La Jolla, USA. [8]Department of Radiation Oncology, University of Alabama Birmingham, Birmingham, AL, USA. [9]Department of Pathology, Emory University, Atlanta, USA. [10]Department of Neurosurgery, Emory University, Atlanta, USA. [11]Department of Pathology, Nationwide Children's Hospital, Columbus, USA. [12]Kaiser Permanente, Atlanta, USA. [13]Neuroscience Institute, Atrium Health, Charlotte, USA. [14]Genitourinary Malignancies Branch, National Cancer Institute, Bethesda, USA. [15]Laboratory of Cancer Biology and Genetics, National Cancer Institute, Bethesda, USA. [16]Department of Medicine, Division of Medical Genetics, University of California San Diego, La Jolla, USA. [17]Department of Microbiology and Immunology, Emory University, Atlanta, USA. [18]These authors contributed equally: Caroline S. Jansen, Meghana S. Pagadala, Maria A. Cardenas. [19]These authors jointly supervised this work: Haydn Kissick and Zachary S. Buchwald. ✉e-mail: haydn.kissick@emory.edu; zbuchwa@emory.edu

