## [Peer Review File · Nature Communications]

Pre-operative stereotactic radiosurgery and peri-operative dexamethasone for resectable brain metastases: a two-arm pilot study evaluating clinical outcomes and immunological correlatesEditorial note: This manuscript has been previously reviewed at another journal that is not operating a transparent peer review scheme. This document only contains reviewer comments and rebuttal letters for versions considered at *Nature Communications*.

REVIEWER COMMENTS

Reviewer #5 (Remarks to the Author):

1. The analyses cohorts in this paper are of small sample size. For example, the pSRS clinical trial only had 21 patients randomized to Arm A and Arm B who were analyzed in this study. Analyses cohorts for exploratory outcomes are also small. Besides the small sample size, the cohorts were also composed of a diverse, heterogenous set of primary tumor sites. Due to the small sample size and heterogeneity in tumor types, comparisons of clinical outcomes or immunologic outcomes between cohorts or comparisons to other previously reported studies could have issues.

Due to the small sample size, it is also hard to evaluate whether different cohorts are comparable in baseline characteristics. For example, for the pSRS clinical trial, Arm B patients appeared to be younger with minimum age of 34.5 (comparing to 49.1 in Arm A) and mean of 57.5 comparing to 63.1 in Arm A).

The paper presented comparisons of cohorts in distribution of patient/disease characteristics. For many variables that were considered, the tables showed p values >0.05 . However, with small sample size, such comparisons do not have sufficient statistical power. A p value >0.05 does not mean there were not meaningful differences.

As a result, the analyses and conclusions presented in this paper could potentially be influenced.

2. Based on Figure 1, the follow-up of the patients appeared short. There appeared to be only 2-3 patients with follow-up data up to 18 months. The median and range of follow-up

of the patients should be provided. It will also be helpful to discuss whether assessment of outcomes/safety of the treatment would need longer follow-up.

3. For the primary/secondary analyses, the analyses were not intent-to-treat and did not include all randomized patients. Among the 26 randomized patients, 21 patients were included in the analyses. What happened to the other 5 patients? What were the reasons that they didn't receive pSRS or surgery? Comparison of outcomes of all randomized patients should be added.

4. For Adverse Events (Table 2), what was the time frame for evaluation of AEs?

5. Since patients did not have very long overall survival, the paper should clearly define time 0 for each time-to-event endpoint (e.g., cumulative incidence analysis in Figure 1, Figure 2, Supp Figure 2, etc., as well as the reported results on OS). In addition, for reported survival rates or for cumulative incidence rates, a measurement of precision should be added (e.g., 95% confidence interval).

6. For Figure 2I, how was the cutoff for higher density of immune niches selected? The paper should describe the method for selection of cutoffs and whether the choice of the cutoff was data driven.

Thank you for the feedback on our manuscript revisions. Below is our detailed response.

Reviewer #5 (Remarks to the Author):

1. The analyses cohorts in this paper are of small sample size. For example, the pSRS clinical trial only had 21 patients randomized to Arm A and Arm B who were analyzed in this study. Analyses cohorts for exploratory outcomes are also small. Besides the small sample size, the cohorts were also composed of a diverse, heterogenous set of primary tumor sites. Due to the small sample size and heterogeneity in tumor types, comparisons of clinical outcomes or immunologic outcomes between cohorts or comparisons to other previously reported studies could have issues.

Due to the small sample size, it is also hard to evaluate whether different cohorts are comparable in baseline characteristics. For example, for the pSRS clinical trial, Arm B patients appeared to be younger with minimum age of 34.5 (comparing to 49.1 in Arm A) and mean of 57.5 comparing to 63.1 in Arm A).

The paper presented comparisons of cohorts in distribution of patient/disease characteristics. For many variables that were considered, the tables showed p values >0.05 . However, with small sample size, such comparisons do not have sufficient statistical power. A p value >0.05 does not mean there were not meaningful differences.

As a result, the analyses and conclusions presented in this paper could potentially be influenced.

Thank you for the feedback. We agree and therefore we do not make any firm conclusions in the results section regarding the clinical implications and outcomes of the prospective trial. We have added additional qualifying statements in the discussion.

“The clinical trial had a few limitations. First, long term safety evaluation are needed to draw more extensive clinical conclusions. Second, it was not powered for secondary outcomes including comparing the low and high dose steroid Arms. Therefore, although no significant differences in outcomes were observed, validation on a larger trial will be required.”

2. Based on Figure 1, the follow-up of the patients appeared short. There appeared to be only 2-3 patients with follow-up data up to 18 months. The median and range of follow-up of the patients should be provided. It will also be helpful to discuss whether assessment of outcomes/safety of the treatment would need longer follow-up.

Follow-up information has been included.

The median follow-up was 10.32 months (range: 0.84-23.04; interquartile range: 11.76) for n=21.

Discussion of the outcomes/safety is noted in response to critique #1.

3. For the primary/secondary analyses, the analyses were not intent-to-treat and did not include all randomized patients. Among the 26 randomized patients, 21 patients were included in the analyses. What happened to the other 5 patients? What were the reasons that they didn't receive pSRS or surgery? Comparison of outcomes of all randomized patients should be added.

Below are the data for 23 patients with data similar to the per-protocol analysis. As the pre-specified starting point for outcomes analysis was the delivery of SRS, analysis was not performed on the 3 patients that did not receive pSRS. Additionally, the primary endpoint of this study was to assess the toxicity associated with pSRS and surgery, and therefore the analysis was done per protocol.

Reason for not receiving pre-operative radiation: 1 patient converted to whole brain due to rapid intracranial progression; 1 declined radiation after enrolling; 1 signed a revocation letter prior to radiation administration.

Reason for not receiving surgery: 1 passed away due to a cardiac event, 1 due to systemic disease progression.

4. For Adverse Events (Table 2), what was the time frame for evaluation of AEs?

Events that occurred up to 4 months post pSRS. This was the pre-specified primary endpoint for AE evaluation and reporting.

5. Since patients did not have very long overall survival, the paper should clearly define time 0 for each time-to-event endpoint (e.g., cumulative incidence analysis in Figure 1, Figure 2, Supp Figure 2, etc., as well as the reported results on OS). In addition, for reported survival rates or for cumulative incidence rates, a measurement of precision should be added (e.g., 95% confidence interval).

Day 0 of all time-to-event endpoints is date of pre-operative stereotactic radiosurgery. 95% CIs added in the text. This has been added in the methods section.

6. For Figure 2I, how was the cutoff for higher density of immune niches selected? The paper should describe the method for selection of cutoffs and whether the choice of the cutoff was data driven.

Yes, the selection of cutoffs was data driven. The optimal cut-off value for immune niches was identified by evaluating all possible unique cut-off points. This value was determined by finding the point that maximizes separation in survival analysis while minimizing the bias-adjusted log-rank p-value. We have mentioned in the statistical analysis section that optimal cutoff values were determined by bias-adjusted log-rank test. Additionally, we have now provided the reference for the statistical approach in the manuscript.

"Statistical Analysis: The optimal cutoff values for stem-like T cells and immune niche proportion was determined by bias-adjusted log-rank test (ref)"

Ref: Contal C, O'Quigley J. An application of changepoint methods in studying the effect of age on survival in breast cancer. *Comp Stat Data Anal.* 1999;30:253–270.
doi: 10.1016/S0167-9473(98)00096-6.